# Alcohol use recording in adults with depression in English primary care: a cross-sectional study

Elizabeth Adesanya ![ORCID],[1] Sarah Cook,[1,2] Elizabeth Crellin,[3] Sinead Langan ![ORCID],[1] Kathryn Mansfield ![ORCID],[1] Liam Smeeth,[1] Emily Herrett[1]

¹Department of Non-Communicable Disease Epidemiology, London School of Hygiene & Tropical Medicine, London, UK
²National Heart and Lung Institute, Imperial College London, London, UK
³The Health Foundation, London, UK

**Correspondence to**
Elizabeth Adesanya;
elizabeth.adesanya@lshtm.ac.uk

## ABSTRACT

**Objectives** To investigate alcohol use recording in people with newly diagnosed depression in English primary care and individual characteristics associated with the recording of alcohol use.

**Design** A population-based cross-sectional study.

**Setting** Primary care data from English practices contributing to the UK Clinical Practice Research Datalink.

**Participants** We included adults (18+ years) diagnosed with depression between 1 January 2011 and 1 January 2017 without previous antidepressant use and at least 1 year of registration before diagnosis.

**Primary and secondary outcome measures** We described the proportion of individuals with alcohol use and level of alcohol use recorded at four time points (the date of depression diagnosis, 3 months before or after depression diagnosis, 12 months before or after depression diagnosis and any point pre or postdepression diagnosis). We used logistic regression to investigate individual characteristics associated with alcohol use recording in the 3 months before or after depression diagnosis.

**Results** We identified 36 424 adults with depression. 538 (2%) had alcohol use recorded in the 3 months before or after depression diagnosis using formal validated methods such as the Alcohol Use Disorders Identification Test and its abbreviated versions. At each time point, most individuals with alcohol use recorded were low risk drinkers. Alcohol use recording in the 3 months before or after depression diagnosis was associated with male sex (OR=1.38, 95% CI 1.29 to 1.48) and several other individual-level factors.

**Conclusions** Our study shows low levels of alcohol use recording in the 3 months before or after depression diagnosis. Levels of alcohol use recording varied depending on individual characteristics. Incentivised recording of alcohol use will increase completeness, which could improve clinical management and reduce missed opportunities for care in people with depression.

## INTRODUCTION

Harmful alcohol use is a leading risk factor for premature death and disability.[1] In England, 22% of adults consume harmful levels of alcohol,[2] with alcohol-related harm costing the National Health Service £3.5 billion annually.[3] Harmful alcohol use

### Strengths and limitations of this study

► This study used Clinical Practice Research Datalink data that is broadly representative of the UK population to provide generalisable, real-world evidence of alcohol use recording in people with depression in English primary care.

► We used formal (ie, the Alcohol Use Disorders Identification Test) and informal (ie, units consumed per week) methods to identify and provide detailed descriptions of alcohol use in people with depression.

► Primary care records may not completely capture alcohol screening and alcohol usage, so our findings relating to recorded alcohol use may, therefore, underestimate both levels of alcohol screening and alcohol use among individuals with depression.

► Differences in coding practices between individual general practitioners (GPs) and between different GP practices, and ambiguity in descriptions of alcohol use levels (eg, whether they capture actual alcohol consumption in the previous week or typical weekly consumption) may have led to misclassification of alcohol use where it is recorded.

has been linked with the common mental health condition of depression.[4] In 2014, 20% of people aged 16 and over in the UK had symptoms of anxiety or depression.[5] The relationship between harmful alcohol use and depression is bidirectional—people with depression are more likely to drink harmfully, and, harmful levels of alcohol can act as a depressant, leading people who drink to excess experiencing symptoms of depression.[6,7] In people with depression, harmful alcohol consumption may exacerbate depression through several mechanisms: (1) alcohol can affect the adherence and effectiveness of depression treatment, worsening treatment outcomes[8]; (2) evidence suggests that heavy alcohol use can directly lower serotonin levels (a hormone hypothesised to play a role in depression pathophysiology)[9,10] and (3) harmful alcohol use may worsen depressive

symptoms by intensifying feelings of shame, guilt and low self-esteem, all of which are symptoms of depression.[9]

Screening for alcohol use allows early detection of harmful alcohol use and subsequent intervention before health issues become pronounced or irreversible.[11] UK National Institute for Health and Care Excellence (NICE) guidelines published in 2010 emphasise alcohol screening in primary care for subgroups at increased risk of alcohol abuse, including people with depression.[12] The Alcohol Use Disorders Identification Test (AUDIT) is a commonly used screening tool, and abbreviated versions have been developed for rapid screening in primary care and emergency departments (Fast Alcohol Screening Test (FAST), AUDIT for consumption (AUDIT-C)).[13] Alcohol use is also recorded in primary care health records using other measures such as drinking status and level of alcohol consumption.[14 15]

Given the burden of harmful alcohol use and depression in the UK, and the damaging effects of harmful alcohol use in people with depression, it is crucial that we understand the extent to which alcohol use is recorded in people with depression. Previous work has examined alcohol use recording generally in UK primary care,[14–16] but this is the first study to investigate alcohol use recording specifically in people with newly diagnosed depression.

## METHODS
### Study design and setting
We identified adults (18+ years) with depression using prospectively collected primary care electronic health record data from the UK Clinical Practice Research Datalink (CPRD) GOLD database[17] and linked Index of Multiple Deprivation (IMD) data.[18] The CPRD is an ongoing, nationwide primary care database of routinely collected and anonymised medical records.[17] We included all adults (≥18 years) in the CPRD GOLD database registered with a CPRD practice in England with a first depression diagnosis recorded between 1 January 2011 and 1 January 2017. We excluded individuals if they had less than 1 year of research standard registration prior to their depression diagnosis, or they had previously been prescribed antidepressants (to better capture newly diagnosed depression).

### Factors associated with recording of alcohol use
We investigated a range of potential individual-level factors that could be associated with alcohol use recording in primary care for people with depression: age at depression diagnosis, sex, ethnicity, deprivation, geographical region, smoking status, body mass index (BMI), the number of years after general practitioner (GP) registration that depression was diagnosed (1–2 years, 2–3 years, >3 years), depression management with medication (antidepressant classes: tricyclic antidepressants, selective serotonin uptake inhibitors), monoamine oxidase inhibitors and other antidepressants) or without

medication (talking therapies, eg, cognitive behavioural therapy) and selected comorbidities related to alcohol use (liver disease, hypertension, diabetes mellitus, anxiety and substance abuse).

We defined deprivation using quintiles of the IMD. We identified liver disease, hypertension and diabetes mellitus based on the presence of a recorded (morbidity-coded) diagnosis in primary care prior to the date of the first depression diagnosis. We identified anxiety and substance abuse based on the presence of a (morbidity coded) diagnosis up to 5 years prior to the date of the first depression diagnosis. We used existing code lists and algorithms to define BMI, ethnicity and smoking status.[19 20] For BMI and smoking status, we used the status recorded closest to the date of the first depression diagnosis. We identified depression management with medication using primary care prescription data recorded after the first ever depression diagnosis. We identified talking therapies based on the presence of morbidity codes for psychological therapies (eg, cognitive behavioural therapy, counselling). We have made all code lists used in this study available to download from online repositories.[21 22] Algorithms used to identify BMI and smoking status are described in further detail in the online supplemental material.

### Alcohol use recording
We identified recorded alcohol use using primary care morbidity coding (read codes) and data captured in structured data-entry fields. Records of alcohol use included:
1. Codes indicating the use of AUDIT, AUDIT-C or FAST screening and their associated scores.
2. Codes indicating the diagnosis of alcohol use disorders.
3. Codes quantifying alcohol use:
   – Drinking status (current, non, ex).
   – Drinking level (read coded as light, moderate, heavy or non).
   – Units consumed per week (none, 1–14, 15–42, 43+).

We categorised alcohol use recording in terms of: (1) type: whether it was formal (recorded using formal, validated methods, that is, AUDIT, AUDIT-C or FAST), or informal (ie, morbidity codes suggesting individuals were asked about drinking status or alcohol intake or had an AUD diagnosis) and (2) timing: of recorded alcohol use in relation to depression diagnosis at four time points (date of depression diagnosis; in the 3 months before or after depression diagnosis; in the 12 months before or after depression diagnosis or at any time pre or postdepression diagnosis).

### Statistical analysis
We described the prevalence and level of recorded alcohol use in individuals diagnosed with depression according to whether it was: (1) formal or informal and (2) timing in relation to date of depression diagnosis. Individuals whose alcohol use was recorded using formal, validated screening tools (eg, AUDIT) had their level of alcohol use described according to the recorded screening

tool scores. Individuals whose alcohol use was recorded using informal methods (eg, drinking status or alcohol consumption) had their level of alcohol use based on the recorded information (eg, units per week). Individuals in the study may have been recorded more than once at each time point and using more than one method of alcohol use recording. We used logistic regression to estimate ORs (adjusted for age, sex, ethnicity and deprivation) for the association between the factors potentially associated with alcohol use recording in the 3 months before or after depression diagnosis. Individuals with missing data on any of the variables considered were excluded from regression analyses. Data were analysed using Stata V.15 (StataCorp, Texas, USA).

### Patient and public involvement

Patients or the public were not involved in the design or analysis of this study.

## RESULTS
### Study population

We identified 36 434 adults with a depression diagnosis recorded between 1 January 2011 and 1 January 2017 who were eligible for study inclusion (online supplemental figure 1). Fifty-four per cent (n=19 604) of the study population were women, and the median age was 38 (IQR=27–51). There was an approximately even distribution of the study population across the deprivation quintiles.

### Prevalence of alcohol use recording and level of alcohol use among those recorded

Of 31 007 (85%) individuals had their alcohol use recorded at any point pre or postdepression diagnosis. However, only 6012 (17%) had their alcohol use recorded in the 3 months before or after their depression diagnosis (figure 1).

### Formal alcohol use recording

Alcohol use recording using formal validated methods (AUDIT, AUDIT-C or FAST) was low; 23% (n=8284) of the study population had their alcohol use recorded at any point pre or postdepression diagnosis, and only 2% (n=538) had formal alcohol screening recorded in the 3 months before or after depression diagnosis (table 1). Most individuals who had their alcohol use recorded at each of the time points using AUDIT or FAST were identified as low-risk drinkers (table 1). In the 3 months before or after depression diagnosis, 17% (63/370) of people who had their alcohol use recorded using AUDIT, and 17% (20/116) of those recorded using FAST, were identified as harmful drinkers (ie, score of ≥8 on AUDIT tool and ≥3 on FAST tool). In the 3 months before or after depression diagnosis, 18% (67/370) of those with their alcohol use recorded using AUDIT, and 19% recorded using FAST (22/116), had invalid scores recorded (ie, score of >40 on AUDIT tool and >16 on FAST tool). Over 90% of individuals at each of the four time points

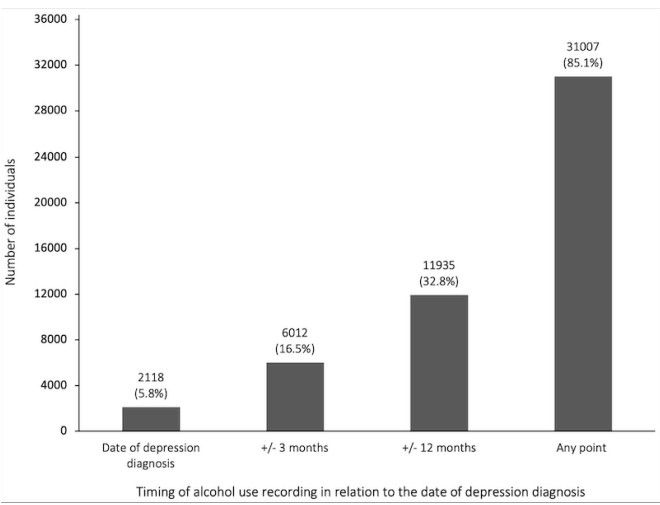

**Figure 1** Prevalence of alcohol use recording at different timepoints in relation to the date of depression diagnosis. **±3 months, Alcohol use recorded in the 3 months before or after depression diagnosis; ±12 months, Alcohol use recorded in the 12 months before or after depression diagnosis; Any point, Alcohol use recorded at any point pre- or post-depression diagnosis.**

who were screened with AUDIT-C did not have a score recorded by their GP.

### Informal alcohol use recording

Sixteen per cent (n=5724) of the study population had their alcohol use recorded using informal methods (ie, codes suggesting that individuals were asked about drinking status or alcohol intake, or had an AUD diagnosed) in the 3 months before or after depression diagnosis, and 83% (n=30 175) of individuals were recorded at any point pre or postdepression diagnosis (table 2). Of the 51% (n=2915) of individuals who had their weekly alcohol consumption recorded (a method of informal alcohol use recording) in the 3 months before or after depression diagnosis, 56% (n=1621) were low-risk drinkers, while 37% (n=1087) were harmful drinkers (defined as consuming more than the recommended 14 units a week for men and women). Twenty-five per cent (n=6818) of individuals who had their weekly alcohol consumption recorded (in units per week) at any point pre or postdepression diagnosis were harmful drinkers.

### Factors associated with alcohol use recording

We saw evidence of increased odds of alcohol use recording using both formal validated methods (AUDIT, AUDIT-C and FAST) and informal methods (drinking level, alcohol consumption, drinking status) in the 3 months before or after depression diagnosis after adjusting for age, sex, ethnicity and deprivation in men, people living in the North of England compared with those living in the South, current or ex-smokers, individuals with their depression managed through talking therapies compared with those that go without and people with hypertension or diabetes (figure 2, online supplemental table 1). There was also evidence of

**Table 1** Number of individuals with alcohol use recorded using formal validated methods (AUDIT, AUDIT-C or FAST) at pre-specified timepoints in relation to depression diagnosis (data are N(%) unless otherwise specified)

| | Timing of alcohol use recording in relation to depression diagnosis N (%) | | | |
|---|---|---|---|---|
| | On date of depression diagnosis | ±3 months | ±12 months | Any point |
| **Formal alcohol use recording (any)** * | **93 (0.3)** | **538 (1.5)** | **1515 (4.2)** | **8284 (22.7)** |
| AUDIT † | 68 (73.1) | 370 (68.8) | 1029 (67.9) | 4615 (55.7) |
| Low risk (AUDIT score 0–7) | 34 (50.0) | 240 (64.9) | 739 (71.8) | 3193 (69.2) |
| Hazardous (AUDIT score 8–15) | 11 (16.2) | 43 (11.6) | 95 (9.2) | 318 (6.9) |
| High risk (AUDIT score 16–19) | ‡ | 7 (1.9) | 10 (1.0) | 27 (0.6) |
| Possible dependence (AUDIT score 20–40) | ‡ | 13 (3.5) | 18 (1.8) | 37 (0.8) |
| *Invalid score value recorded (AUDIT score >40)* | 17 (25.0) | 67 (18.1) | 167 (16.2) | 1040 (22.5) |
| AUDIT-C † | 22 (23.7) | 139 (25.8) | 395 (26.1) | 2886 (34.8) |
| Low risk (AUDIT-C score 0–4) | 0 (0.0) | 10 (7.2) | 22 (5.6) | 79 (2.7) |
| Hazardous (AUDIT-C score 5–7) | 0 (0.0) | ‡ | ‡ | 19 (0.7) |
| High risk (AUDIT-C score 8–10) | 0 (0.0) | 0 (0.0) | ‡ | 6 (0.2) |
| Possible dependence (AUDIT-C score 11–12) | 0 (0.0) | 0 (0.0) | 0 (0.0) | 0 (0.0) |
| *AUDIT-C score not recorded* | 22 (100.00) | 128 (92.1) | 366 (92.7) | 2782 (96.4) |
| FAST † | 16 (17.2) | 116 (21.6) | 368 (24.3) | 2849 (34.4) |
| Low risk (FAST score 0–2) | 11 (68.8) | 74 (63.8) | 242 (65.8) | 1633 (57.3) |
| Hazardous (FAST score 3–16) | ‡ | 20 (17.2) | 48 (13.0) | 336 (11.8) |
| *Invalid score value recorded (FAST score >16)* | ‡ | 22 (19.0) | 78 (21.2) | 880 (30.9) |

Values in bold are the total number of individuals with alcohol use recorded at timepoints using formal validated methods.
*Percentage calculated out of the total number of individuals in the cohort (36,434).
†Percentage calculated out of the total number of individuals with alcohol use recorded using formal validated methods—individuals may have more than one type of formal recording, so numbers add to >100%; risk scores represent the proportion at risk identified using that particular test.
‡Cell counts <5 suppressed to preserve individuals' confidentiality.
Any point, Alcohol use recorded at any point pre- or post-depression diagnosis; AUDIT, Alcohol Use Disorders Identification Test; AUDIT-C, AUDIT for consumption; FAST, Fast Alcohol Screening Test; ±3 months, Alcohol use recorded in the 3 months before or after depression diagnosis; ± 12 months, Alcohol use recorded in the year before or after depression diagnosis.

linear associations between deprivation, time since GP registration and alcohol use recording. The least deprived individuals were more likely to have their alcohol use recorded than the most deprived individuals. Individuals registered at their GP practice for more than 3 years had higher odds of alcohol use recording compared with those registered between 1 and 2 years. There was strong evidence of a non-linear association (p<0.001) between alcohol use recording and age at depression diagnosis. Individuals aged between 60 and 69 had more than two times the odds of alcohol use recording compared with those aged 18–29.

## DISCUSSION
### Summary
We found that levels of alcohol use recording in people with depression were high: 85% of individuals had their

alcohol use recorded using either formal or informal methods at any point pre or postdepression diagnosis. However, only 17% of individuals had their alcohol use recorded in the 3 months before or after depression diagnosis, with only 2% of these individuals recorded using formal validated tools (AUDIT, AUDIT-C or FAST). Just over one-third (37%) of individuals who had their weekly alcohol consumption recorded in the 3 months before or after depression diagnosis were harmful drinkers (defined as consuming more than the recommended 14 units a week for men and women), whereas 25% of individuals who had their weekly alcohol consumption recorded at any point pre or postdepression diagnosis were harmful drinkers.

After adjusting for age, sex, ethnicity and deprivation, we found that several individual-level characteristics

**Table 2** Number of individuals with alcohol use recorded using informal methods (ie, alcohol status, level of alcohol use) at pre-specified timepoints in relation to depression diagnosis (data are N(%) unless otherwise specified)

| | Timing of alcohol use recording in relation to depression diagnosis N (%) | | | |
| --- | --- | --- | --- | --- |
| | Date of depression diagnosis | ±3 months | ±12 months | Any point |
| **Informal alcohol use recording (any)** * | **2039 (5.6)** | **5724 (15.7)** | **11 361 (31.2)** | **30 175 (82.8)** |
| Recording of current alcohol status † | 901 (44.2) | 3025 (52.9) | 6643 (58.5) | 24 195 (80.2) |
| Non | 313 (34.7) | 947 (31.3) | 2081 (31.3) | 8145 (33.7) |
| Current | 496 (55.1) | 1777 (58.7) | 3907 (58.8) | 14 198 (58.7) |
| Ex | 92 (10.2) | 301 (10.0) | 655 (9.9) | 1852 (7.7) |
| Recording of codes indicating level of alcohol use † | 855 (41.9) | 2893 (50.5) | 6406 (56.4) | 24 377 (80.8) |
| None | 405 (47.4) | 1239 (42.8) | 2675 (41.8) | 9370 (38.4) |
| Light | 304 (35.6) | 1260 (43.6) | 2956 (46.1) | 11 383 (46.7) |
| Moderate | 36 (4.2) | 123 (4.3) | 282 (4.4) | 2106 (8.6) |
| Heavy | 110 (12.9) | 271 (9.4) | 493 (7.7) | 1518 (6.2) |
| Recording of units of alcohol consumed per week † | 1090 (53.5) | 2915 (50.9) | 6390 (56.2) | 27 377 (90.7) |
| None | 78 (7.2) | 207 (7.1) | 474 (7.4) | 3744 (13.7) |
| 1–14 | 499 (45.8) | 1621 (55.6) | 3865 (60.5) | 16 815 (61.4) |
| 15–42 | 322 (29.5) | 745 (25.6) | 1498 (23.4) | 5477 (20.0) |
| 43+ | 191 (17.5) | 342 (11.7) | 553 (8.7) | 1341 (4.9) |
| Recording of codes relating to alcohol use disorders † | 39 (1.9) | 117 (2.0) | 213 (1.9) | 584 (1.9) |

Values in bold are the total number of individuals with alcohol use recorded at timepoints using informal methods.
*Percentage calculated out of the total number of individuals in the cohort (36,434).
†Percentage calculated out of the total number of individuals with alcohol use recorded using informal methods—individuals may have more than one type of informal recording, so numbers add to >100%; percentages are out of the number of people with alcohol use recording using that specific method.
Any point, Alcohol use recorded at any point pre- or post-depression diagnosis; ± 3 months, Alcohol use recorded in the 3 months before or after depression diagnosis; ± 12 months, Alcohol use recorded in the year before or after depression diagnosis.

(including living in the least deprived areas, current or ex-smokers and comorbid hypertension or diabetes) were associated with alcohol use recording in the 3 months before or after depression diagnosis.

### Comparison with the existing literature

Compared to a 2019 study, which showed that 52% of people in UK primary care had a record of alcohol use,[14] our study suggests that alcohol use recording in people with depression was higher (85%). Studies using primary care data in the UK and US have also provided evidence that individuals with depression or other psychiatric conditions are more likely to have their alcohol use recorded or screened for an alcohol use disorder compared with the general population.[14 23]

According to data from the 2018 Health Survey for England (HSE), 22% of all adults in England consumed more than the recommended 14 units a week.[2] This is comparable to the figure (25%) in the present study among those with alcohol consumption recorded at any time. However, the proportion of harmful drinkers identified by the HSE is lower than the proportion of individuals with depression in this study who had their alcohol consumption recorded in the 3 months before or after depression diagnosis and were found to be harmful drinkers (37%).

Our findings regarding are comparable to studies in Sweden where the proportion of individuals with depression that consumed harmful amounts of alcohol ranged from 21% to 23% (compared with 25% in our study).[24 25] However, our findings regarding harmful weekly alcohol consumption are dissimilar to studies conducted in: (1) Singapore, where 19% of people with depression were harmful drinkers[26] and (2) South Korea, where 51% of people with depression were harmful drinkers.[27] GP recording practice, as well as cultural and governmental variations, is possible explanations of this difference in results. Lower levels of harmful drinking in Singapore may be explained by strict alcohol laws; alcohol cannot be sold or consumed in public from 22:30 to 7:00.[26] Similarly, the prominent

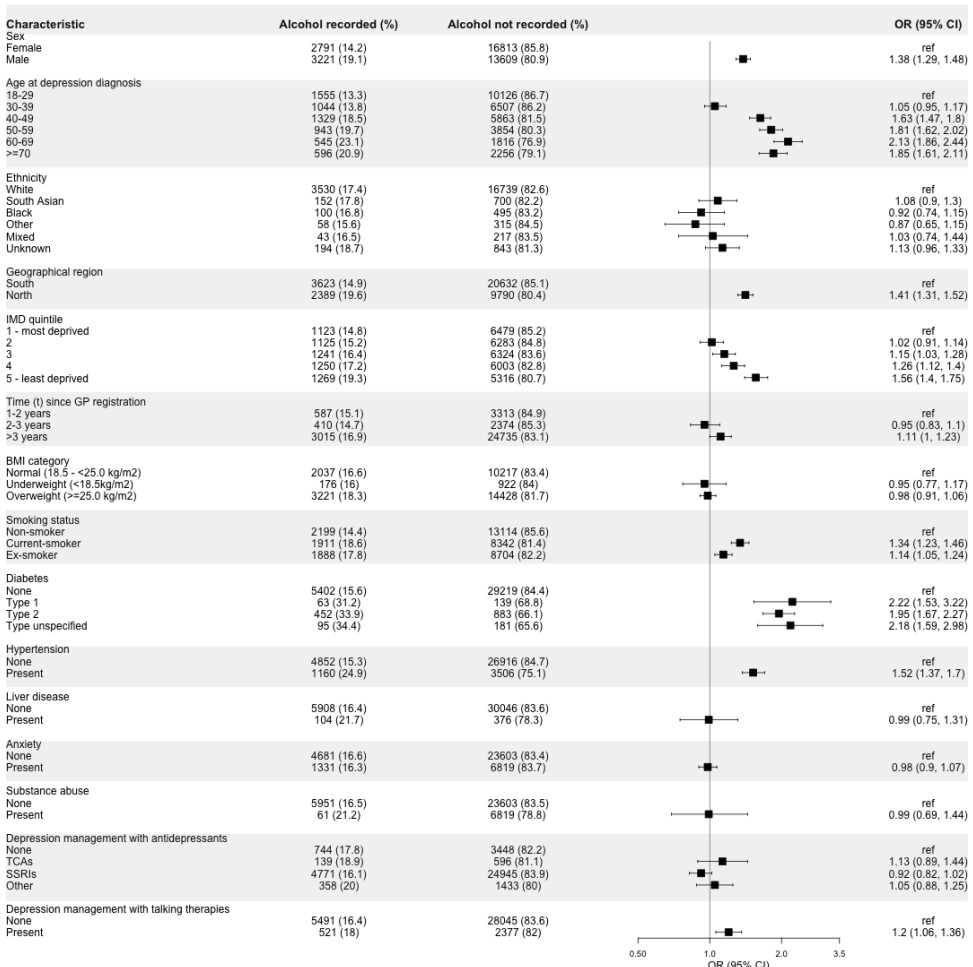

| Characteristic | Alcohol recorded (%) | Alcohol not recorded (%) | OR (95% CI) |
|---|---|---|---|
| **Sex** | | | |
| Female | 2791 (14.2) | 16813 (85.8) | ref |
| Male | 3221 (19.1) | 13609 (80.9) | 1.38 (1.29, 1.48) |
| **Age at depression diagnosis** | | | |
| 18-29 | 1555 (13.3) | 10126 (86.7) | ref |
| 30-39 | 1044 (13.8) | 6507 (86.2) | 1.05 (0.95, 1.17) |
| 40-49 | 1329 (18.5) | 5863 (81.5) | 1.63 (1.47, 1.8) |
| 50-59 | 943 (19.7) | 3854 (80.3) | 1.81 (1.62, 2.02) |
| 60-69 | 545 (23.1) | 1816 (76.9) | 2.13 (1.86, 2.44) |
| >=70 | 596 (20.9) | 2256 (79.1) | 1.85 (1.61, 2.11) |
| **Ethnicity** | | | |
| White | 3530 (17.4) | 16739 (82.6) | ref |
| South Asian | 152 (17.8) | 700 (82.2) | 1.08 (0.9, 1.3) |
| Black | 100 (16.8) | 495 (83.2) | 0.92 (0.74, 1.15) |
| Other | 58 (15.6) | 315 (84.5) | 0.87 (0.65, 1.15) |
| Mixed | 43 (16.5) | 217 (83.5) | 1.03 (0.74, 1.44) |
| Unknown | 194 (18.7) | 843 (81.3) | 1.13 (0.96, 1.33) |
| **Geographical region** | | | |
| South | 3623 (14.9) | 20632 (85.1) | ref |
| North | 2389 (19.6) | 9790 (80.4) | 1.41 (1.31, 1.52) |
| **IMD quintile** | | | |
| 1 - most deprived | 1123 (14.8) | 6479 (85.2) | ref |
| 2 | 1125 (15.2) | 6283 (84.8) | 1.02 (0.91, 1.14) |
| 3 | 1241 (16.4) | 6324 (83.6) | 1.15 (1.03, 1.28) |
| 4 | 1250 (17.2) | 6003 (82.8) | 1.26 (1.12, 1.4) |
| 5 - least deprived | 1269 (19.3) | 5316 (80.7) | 1.56 (1.4, 1.75) |
| **Time (t) since GP registration** | | | |
| 1-2 years | 587 (15.1) | 3313 (84.9) | ref |
| 2-3 years | 410 (14.7) | 2374 (85.3) | 0.95 (0.83, 1.1) |
| >3 years | 3015 (16.9) | 24735 (83.1) | 1.11 (1, 1.23) |
| **BMI category** | | | |
| Normal (18.5 - <25.0 kg/m2) | 2037 (16.6) | 10217 (83.4) | ref |
| Underweight (<18.5kg/m2) | 176 (16) | 922 (84) | 0.95 (0.77, 1.17) |
| Overweight (>=25.0 kg/m2) | 3221 (18.3) | 14428 (81.7) | 0.98 (0.91, 1.06) |
| **Smoking status** | | | |
| Non-smoker | 2199 (14.4) | 13114 (85.6) | ref |
| Current-smoker | 1911 (18.6) | 8342 (81.4) | 1.34 (1.23, 1.46) |
| Ex-smoker | 1888 (17.8) | 8704 (82.2) | 1.14 (1.05, 1.24) |
| **Diabetes** | | | |
| None | 5402 (15.6) | 29219 (84.4) | ref |
| Type 1 | 63 (31.2) | 139 (68.8) | 2.22 (1.53, 3.22) |
| Type 2 | 452 (33.9) | 883 (66.1) | 1.95 (1.67, 2.27) |
| Type unspecified | 95 (34.4) | 181 (65.6) | 2.18 (1.59, 2.98) |
| **Hypertension** | | | |
| None | 4852 (15.3) | 26916 (84.7) | ref |
| Present | 1160 (24.9) | 3506 (75.1) | 1.52 (1.37, 1.7) |
| **Liver disease** | | | |
| None | 5908 (16.4) | 30046 (83.6) | ref |
| Present | 104 (21.7) | 376 (78.3) | 0.99 (0.75, 1.31) |
| **Anxiety** | | | |
| None | 4681 (16.6) | 23603 (83.4) | ref |
| Present | 1331 (16.3) | 6819 (83.7) | 0.98 (0.9, 1.07) |
| **Substance abuse** | | | |
| None | 5951 (16.5) | 23603 (83.5) | ref |
| Present | 61 (21.2) | 6819 (78.8) | 0.99 (0.69, 1.44) |
| **Depression management with antidepressants** | | | |
| None | 744 (17.8) | 3448 (82.2) | ref |
| TCAs | 139 (18.9) | 596 (81.1) | 1.13 (0.89, 1.44) |
| SSRIs | 4771 (16.1) | 24945 (83.9) | 0.92 (0.82, 1.02) |
| Other | 358 (20) | 1433 (80) | 1.05 (0.88, 1.25) |
| **Depression management with talking therapies** | | | |
| None | 5491 (16.4) | 28045 (83.6) | ref |
| Present | 521 (18) | 2377 (82) | 1.2 (1.06, 1.36) |

0.50    1.0    2.0    3.5
OR (95% CI)

**Figure 2** Individual characteristics of alcohol use recording in the 3 months before or after depression diagnosis. BMI, body mass index; GP, General Practitioner; IMD, Index of Multiple Deprivation; OR, Odds Ratio; SSRI, selective serotonin uptake inhibitor; TCA, tricyclic antidepressant.

drinking culture in South Korea may contribute to the increased prevalence.[28]

Our study found that people with depression who were men, ex/current smokers, aged 60–69, diagnosed with diabetes or hypertension, from areas with the lowest level of deprivation, or who lived in the North of England were more likely to have their alcohol use recorded compared with other people with depression. Other UK primary care studies of alcohol use recording have also demonstrated increased rates of alcohol use recording in people with diabetes and hypertension, and those aged 60–69 years.[14–16] There is also evidence to suggest that some of the factors we identified as being linked to alcohol use recording are linked to harmful alcohol use. For example, smokers and ex-smokers have been shown to be more likely to consume harmful amounts of alcohol when compared with non-smokers.[29] Data from the 2018 HSE also showed that men were 15% more likely to drink harmfully compared with women, and the highest proportion of heavier drinkers was found in the North of England.[2] This evidence from HSE taken alongside our results may suggest that GPs are more likely to record

alcohol use in those perceived to be at higher risk even among those with depression.

However, some characteristics have been identified as being associated with increased levels of alcohol use recording in our study contrast with finding from other UK studies. Other studies have shown that women are more likely to have their alcohol use recorded compared with men,[15] and those in the most deprived regions are more likely to be have recorded alcohol use than those in the least deprived areas.[14 15] However, the increased odds of alcohol use recording in the least deprived areas seen in this study may be explained by GP practices in more deprived areas possibly having fewer resources to deliver timely and formal alcohol use recording.

## Strengths and limitations

The CPRD GOLD data set is broadly representative of the UK population with regards to age and sex,[15] and, consequently, our results can be broadly generalised to the rest of the UK population. Our study was able to use complete data on key variables such as age, sex, deprivation and comorbid conditions. Importantly, this study captured

levels of alcohol use recording in a real-life setting in UK primary care, and, therefore, highlights potential missed opportunities for care in people with depression.

An important limitation of this study is that individuals without alcohol use recorded may have discussed alcohol with their GP, but this was not documented, leading to underestimation of alcohol use screening based on recording as a proxy marker. Lack of documentation may be more likely where low levels of drinking were reported. While we cannot directly extrapolate our findings to discussion of alcohol use within GP consultations, our findings related to recording are important given documentation of alcohol use is crucial for informing future management and monitoring change in drinking.

Similarly, although our analysis found an association between various factors and alcohol use recording, GPs may record alcohol use in individuals who they believe are at higher risk of harmful alcohol use for reasons that are not in their coded medical record. GPs may also base their decisions to ask about, and record, alcohol use on the previous alcohol level recorded. For instance, people who had previously had high levels of alcohol use recorded may be more likely to have their alcohol use recorded when they are diagnosed with depression. Equally the decision to record alcohol use if discussed in the consultation may also be related to the level of alcohol use (GPs may be more likely to document harmful drinking), which could influence the estimates of harmful drinking among those with depression found here.

Another limitation is the potential imprecision of alcohol consumption data recorded in primary care. A number of factors related to recording mean that our estimates may be an unreliable estimate of true alcohol intake. First, we noted that some people with alcohol use recorded using formal methods either did not have a score for the test recorded (particularly frequent in those with alcohol use recorded using AUDIT-C), or the recorded score was outside the total possible value for the test. It is unclear how such missing data would affect our findings. Second, it is unclear whether individuals with recorded alcohol use were asked about recent consumption (how much they actually consumed in the past week) or typical consumption (how much they typically consume in a week). The potential ambiguity in the consumption data implies differences in coding practices between individual GPs and between different GP practices. It also suggests that consumption records need to be used cautiously for research as they may not necessarily be capturing the same thing for each individual. And finally, individuals may be less truthful about their alcohol consumption if asked face-to-face when compared with a self-completed questionnaire (potentially underestimating harmful drinking levels).

Data on variables such as substance abuse and depression management without medication are likely to be poorly captured within the CPRD. Restricting entry into the study to individuals without previous antidepressant prescriptions allowed us to more cleanly capture people

with newly diagnosed depression, but it also reduced sensitivity and we may have missed individuals that had depression who had been prescribed antidepressants before depression diagnosis was coded.

## Implication for research and/or practice

Our findings have important implications for alcohol use recording among people with depression in English primary care. Our results indicate that GPs are not recording alcohol use in people with depression as effectively as they could be (in terms of capturing all individuals or using the most appropriate tools). A lack of recording does not mean that appropriate alcohol use screening is not happening in practice. However, the low levels of alcohol use screening recording we saw may suggest that NICE guidelines[12] (recommending alcohol use screening in people with depression) have not been closely followed in UK primary care, suggesting that alcohol use may not have been discussed and addressed by the GP in a timely manner. Not screening for alcohol use represents a potential missed opportunity for care in people with depression that may lead to poorer health outcomes and may highlight an area for improvement within primary care.

While it is encouraging that some alcohol use recording is taking place, it is imperative that all individuals with depression or those at high risk of alcohol misuse have their alcohol use routinely screened in primary care using quick validated tools such as the AUDIT-C or FAST. While these questionnaires are builtin to most GP clinical systems in the UK, more effective and efficient alcohol use screening might be possible through self-administered questionnaires (simply administered in GP waiting rooms or using online tools). Routine screening leads to early detection and the opportunity for GPs to provide brief alcohol interventions that have been proven to be effective at reducing alcohol consumption among people who are drinking at harmful levels.[30]

Evidence shows that financial incentives can increase primary care alcohol use recording.[30] The UK government recently introduced an, 'Alcohol-related risk reduction scheme' requiring GPs to identify newly registered individuals aged 16 or over who drink harmful levels of alcohol; however, there is no emphasis on depression nor funding attached.[31] There is also currently no financial incentive for recording alcohol use in people with depression through the Quality and Outcomes Framework (QOF),[32] even though in 2019, a NICE suggested a QOF indicator to record the percentage of individuals diagnosed with depression in the last year who had been screened for hazardous drinking in the 3 months before or after their depression diagnosis.[33] Additionally, the small financial reward offered to practices in England for screening newly registered adults for harmful alcohol use as part of Clinical Directed Enhanced Services ceased in 2015.[34 35]

The incentivisation of alcohol use recording may help to increase number of people recorded, which is crucial for future patient management and will improve holistic

care for individuals with depression. Adding alcohol use recording as a QOF indicator or rewarding services for recording of alcohol use is likely to improve clinical practice and patient outcomes.

**Acknowledgements** This study is based in part on data from the Clinical Practice Research Datalink obtained under licence from the UK Medicines and Healthcare products Regulatory Agency. The data are provided by patients and collected by the NHS as part of their care and support. The interpretation and conclusions contained in this study are those of the authors alone.

**Contributors** EH, SC and KM designed the study. EC and EH extracted the data. EA performed the analysis and wrote the first draft of the manuscript. All authors (EA, SC, EC, SL, KM, LS and EH) reviewed and approved the final manuscript. EH acts as a guarantor and accepts full responsibility for the finished work and the conduct of the study, had access to the data, and controlled the decision to publish. Dr Alasdair Henderson contributed to the development of Figure 2 in the final manuscript.

**Funding** EH is funded by a post-doctoral fellowship from the NIHR (Reference: PDF-2016-09-029). SML is funded by a Wellcome Trust Senior Clinical Fellowship in Science (Reference: 205039/Z/16/Z). For the purpose of Open Access, the author has applied a CC BY public copyright licence to any Author Accepted Manuscript (AAM) version arising from this submission.

**Competing interests** None declared.

**Patient consent for publication** Not applicable.

**Ethics approval** The study was approved by the London School of Hygiene and Tropical Medicine Research Ethics Committee and by the Clinical Practice Research Datalink Independent Scientific Advisory Committee (ISAC Protocol Number: 19_077).

**Provenance and peer review** Not commissioned; externally peer reviewed.

**Data availability statement** Data may be obtained from a third party and are not publicly available. Data from this study were obtained from the Clinical Practice Research Datalink (CPRD) and cannot be shared directly by researchers. Data are available directly from CPRD subject to independent approval.

**ORCID iDs**
Elizabeth Adesanya http://orcid.org/0000-0002-8912-7520
Sinead Langan http://orcid.org/0000-0002-7022-7441
Kathryn Mansfield http://orcid.org/0000-0002-2551-410X

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
