## [Reviewer comments · BMJ Open]

ARTICLE DETAILS

TITLE (PROVISIONAL)	Alcohol use recording in adults with depression in English primary care: a cross-sectional study
AUTHORS	Adesanya, Elizabeth; Cook, Sarah; Crellin, Elizabeth; Langan, Sinead; Mansfield, Kathryn; Smeeth, Liam; Herrett, Emily

VERSION 1 – REVIEW

REVIEWER	Farren, Conor University of Dublin Trinity College, School of Medicine, Discipline of Psychiatry
REVIEW RETURNED	25-Aug-2021

GENERAL COMMENTS	This is a well written and clear manuscript investigating alcohol use in adults with depression in English primary care. The abstract and the article summary are appropriate. The introduction is brief and appropriate. The authors could add to the justification of the need for the paper, as this is not overwhelming. The methods section is clear, with the individual parameters explicitly detailed. Statistical analysis appears appropriate. Table 1 in the results section did not appear to be reproduced correctly in my copy, and the +-12 months section appeared cut off. Thus I could only go by the description in the results section itself. There appeared to be a considerable redundancy between the information in table 1 and the written results section. Similarly for Table 2, there was some redundancy between the results section and table 2 (which was reproduced fully). For table 3, there appeared to be a considerable overlap in the information with the results section, and table 3 appeared to be overly detailed. The discussion summary appeared to reinforce the findings from the results section, in a somewhat repetitive fashion. The comparison with existing literature section was interesting and added information, such as further international comparisons, would be welcomed. The limitations are detailed sufficiently. The strengths and the implications for research and practice are important and if anything under-emphasised and might benefit from an additional strengthening final statement.
--

REVIEWER	Subramaniam, Mythily Institute of Mental Health, Research
REVIEW RETURNED	07-Oct-2021

GENERAL COMMENTS	The authors must be commended for this interesting, well-designed and extremely relevant piece of research. I have one minor comment for their consideration. The authors may want to elaborate leveraging on alerts to the GPS when a person meets criteria for depression combined with the use
--

	of very short questionnaires assessing harmful alcohol use that can be self-administered by patients using hand held devices.
--	---

VERSION 1 – AUTHOR RESPONSE

Reviewer 1

COMMENT 1.1

This is a well written and clear manuscript investigating alcohol use in adults with depression in English primary care. The abstract and the article summary are appropriate. The introduction is brief and appropriate. The authors could add to the justification of the need for the paper, as this is not overwhelming.

RESPONSE 1.1

Thank you for your kind comments. We have added some additional text to the Introduction of the manuscript to further justify the importance of our study.

MANUSCRIPT CHANGES 1.1

Location – Introduction, page 4

In people with depression, harmful alcohol consumption may exacerbate depression through mechanisms including: 1) alcohol affecting the adherence to, and effectiveness of, depression treatment worsening depression outcomes;⁸ 2) evidence suggests heavy alcohol use can directly lower serotonin levels (a hormone with a hypothesised role in depression pathophysiology);^{9,10} and 3) harmful alcohol use may worsen depressive symptoms by intensifying feelings of shame, guilt and low self-esteem, all of which are symptoms of depression.⁹

COMMENT 1.2

The methods section is clear, with the individual parameters explicitly detailed. Statistical analysis appears appropriate. Table 1 in the results section did not appear to be reproduced correctly in my copy, and the +12 months section appeared cut off. Thus I could only go by the description in the results section itself. There appeared to be a considerable redundancy between the information in table 1 and the written results section. Similarly for Table 2, there was some redundancy between the results section and table 2 (which was reproduced fully). For table 3, there appeared to be a considerable overlap in the information with the results section, and table 3 appeared to be overly detailed.

RESPONSE 1.2

Thank you for your comments. We disagree that the descriptions of Tables 1 and 2 in the written Results section are redundant. We believe that the corresponding text in the Results section for Tables 1 and 2 provides a clear and easy narrative explanation of two complex tables. However, if the editors think that it is important, we are happy to edit the text down.

For Table 3, we agree that there is considerable overlap in the information within the table and the written Results section, and that the table is overly detailed. To take these points into consideration, we have more succinctly summarised the results of Table 3 in the corresponding text and presented the key results from Table 3 in a forest plot (**Figure 2**). Table 3 is now included as part of the Supplementary material (**Supplementary Table 1**).

MANUSCRIPT CHANGES 1.2

Location – Results: Factors associated with alcohol use recording, page 8

We saw evidence of increased odds of alcohol use recording using both formal validated methods (AUDIT, AUDIT-C, FAST) and informal methods (drinking level, alcohol consumption, drinking status) in the three months before or after depression diagnosis after adjusting for age, sex, ethnicity, and deprivation in: men, people living in the North of England compared to those living in the South, current or ex-smokers, individuals with their depression managed through talking therapies compared to those that go without, and people with hypertension or diabetes (**Figure 2, Supplementary Table 1**). There was also evidence of linear associations between deprivation, time since practice registration and alcohol use recording. The least deprived individuals were more likely to have their alcohol use recorded than the most deprived individuals. Individuals registered at their GP practice for more than three years had higher odds of alcohol use recording compared to those registered between

1-2 years. There was strong evidence of a non-linear association ($p < 0.001$) between alcohol use recording and age at depression diagnosis. Individuals aged between 60-69 had more than twice the odds of alcohol use recording compared to those aged 18-29.

Location – Figure 2: Characteristics and odds of alcohol use recording in the three months before or after depression diagnosis

COMMENT 1.3

The discussion summary appeared to reinforce the findings from the results section, in a somewhat repetitive fashion. The comparison with existing literature section was interesting and added information, such as further international comparisons, would be welcomed. The limitations are detailed sufficiently. The strengths and the implications for research and practice are important and if anything under-emphasised and might benefit from an additional strengthening final statement.

RESPONSE 1.3

We have edited the initial summary section in the Discussion to avoid repetition with the Results section. We have also added text to further compare our results to those from other international studies. Finally, we have included additional text to the 'Implications for research and/or practice' section in the Discussion to further emphasise what our results may mean clinically and for policy overall.

MANUSCRIPT CHANGES 1.3

Location – Discussion: Summary, page 13

After adjusting for age, sex, ethnicity, and deprivation, we found that several individual-level characteristics (including living in the least deprived areas, current or ex-smokers, and comorbid hypertension or diabetes) were associated with alcohol-use recording in the three months before or after depression diagnosis.

Location – Discussion: Comparison with existing literature, page 13

Studies using primary care data in the UK and US have also provided evidence that individuals with depression or other psychiatric conditions are more likely to have their alcohol use recorded or be screened for an alcohol use disorder compared to the general population.^{14,23}

Our findings regarding are comparable to studies in Sweden where the proportion of individuals with depression that consumed harmful amounts of alcohol ranged from 21-23% (compared to 25% in our study).^{24,25} However, our findings regarding harmful weekly alcohol consumption are dissimilar to studies conducted in: 1) Singapore, where 19% of people with depression were harmful drinkers;²⁶ and 2) South Korea, where 51% of people with depression were harmful drinkers.²⁷ GP recording practice, as well as cultural and governmental variations, are possible explanations of this difference in results.

Location – Discussion: Implications for research and/or practice, page 16

Our findings have important implications for alcohol use recording among people with depression in English primary care. Our results indicate that GPs are not recording alcohol use in people with depression as effectively as they could be (in terms of capturing all individuals or using the most appropriate tools). A lack of recording does not mean that appropriate alcohol-use screening is not happening in practice. However, the low levels of alcohol-use screening recording we saw may suggest that NICE guidelines¹² (recommending alcohol-use screening in people with depression) have not been closely followed in UK primary care, suggesting that alcohol use may not have been discussed and addressed by the GP in a timely manner. Not screening for alcohol-use represents a potential missed opportunity for care in people with depression that may lead to poorer health outcomes and may highlight an area for improvement within primary care.

While it is encouraging that some alcohol use recording is taking place, it is imperative that all individuals with depression or those at high risk of alcohol misuse have their alcohol use routinely screened in primary care using quick validated tools such as the AUDIT-C or FAST. While these questionnaires are built-in to most GP clinical systems in the UK, more

effective and efficient alcohol-use screening might be possible through self-administered questionnaires (simply administered in GP waiting rooms or using online tools). Routine screening leads to early detection and the opportunity for GPs to provide brief alcohol interventions that have been proven to be effective at reducing alcohol consumption among people who are drinking at harmful levels.³⁰

Evidence shows that financial incentives can increase primary care alcohol use recording.³⁰ The UK government recently introduced an, 'Alcohol related risk reduction scheme' requiring GPs to identify newly registered individuals aged 16 or over who drink harmful levels of alcohol, however there is no emphasis on depression nor funding attached.³¹ There is also currently no financial incentive for recording alcohol use in people with depression through the Quality and Outcomes Framework (QOF),³² even though in 2019, a NICE suggested a QOF indicator to record the percentage of individuals diagnosed with depression in the last year who had been screened for hazardous drinking in the three months before or after their depression diagnosis.³³ Additionally, the small financial reward offered to practices in England for screening newly registered adults for harmful alcohol use as part of Clinical Directed Enhanced Services (DES) ceased in 2015.^{34,35}

Reviewer 2

COMMENT 2.1

The authors must be commended for this interesting, well-designed and extremely relevant piece of research. I have one minor comment for their consideration. The authors may want to elaborate leveraging on alerts to the GPs when a person meets criteria for depression combined with the use of very short questionnaires assessing harmful alcohol use that can be self-administered by patients using hand held devices.

RESPONSE 2.1

Thank you for your kind comments. We agree that we should offer examples of how alcohol use screening can be improved in primary care and we have provided suggestions within the text.

MANUSCRIPT CHANGES 2.1

Location – Discussion: Implications for research and/or practice, page 16

See our manuscript edits in response to Reviewer 1, Comment 3 (**MANUSCRIPT CHANGES 1.3**)